# WSSe Nanocomposites for Enhanced Photocatalytic Hydrogen Evolution and Methylene Blue Removal under Visible-Light Irradiation

**DOI:** 10.3390/ma15165616

**Published:** 2022-08-16

**Authors:** Tsung-Mo Tien, Yu-Jen Chung, Chen-Tang Huang, Edward L. Chen

**Affiliations:** 1Coastal Water and Environment Center, College of Hydrosphere Science, National Kaohsiung University of Science and Technology, Kaohsiung City 81157, Taiwan; 2Department of Marine Science, Republic of China Naval Academy, Kaohsiung 81300, Taiwan

**Keywords:** nanocomposite, hydrogen production, photocatalytic, Methylene blue removal

## Abstract

In this study, a novel tungsten disulfide diselenide (WSSe) nanocomposite by a facile hydrothermal process with great capable photocatalytic efficiency for hydrogen evolution from water and organic compound removal was discussed. The WSSe nanocomposites form heterojunctions in order to inhibit the quick recombination rate of photo-induced electrons and holes. This is considered to be a useful method in order to enhance the capability of photocatalytic hydrogen production. The hydrogen production rate of the WSSe nanocomposites approaches 3647.4 μmol/g/h, which is 12 and 11 folds the rates of the bare WS_2_ and WSe_2_, respectively. Moreover, the excellent photocatalytic performance for Methylene blue (MB) removal (88%) was 2.5 and 1.8 times higher than those of the bare WS_2_ and WSe_2_, respectively. The great photocatalytic efficiency was owing to the capable electrons and holes separation of WSSe and the construction of the heterostructure, which possessed vigorous photocatalytic oxidation and reduction potentials. The novel one-dimensional structure of the WSSe heterojunction shortens the transport pathway of the photo-induced electrons and holes. It possesses the great capable photocatalytic efficiency of the hydrogen production and organic dye removal. This study offers an insight into the route of interfacial migration and separation for induced charge carriers in order to generate clean hydrogen energy and to solve the issue of environmental pollution.

## 1. Introduction

Owing to the consecutive development of broad industrialization, the global economy has made significant progress. As a powerful and environmentally friendly system for hydrogen evolution, photocatalytic technology was studied as a promising solution to these problems [1,2]. Solar energy has become one of the most attractive renewable energy sources due to its capability to be reused, it is pollution free, and is highly clean [3,4,5]. Therefore, the great photo-induced capability, the lack of secondary pollution, and the low-cost process for decomposing water (H_2_O) into hydrogen (H_2_) are the highlights of the present research [6,7,8]. In addition, a favored photocatalytic hydrogen evolution route that is essential and wide, and has a vigorous light absorption in order to generate a greater amount of charge carriers, and has a useful electrons and holes separation efficiency as well as proper energy band gaps in order to induce the response of reducing H_2_O to H_2_.

In photocatalysis, the primary points affecting the photocatalytic capability include the visible light application potential, the migration, and separation performance of photogenerated electron and hole pairs [9,10,11]. It has been noted that the bulk of photocatalysts are critical for a single component in order for it to have a greater photocatalytic capability that would benefit its function because of its confined high recombination rate of photoexcited electron and hole pairs and light absorption efficiency. Heterojunction photocatalytic activities can apply renewable solar light energy in order to produce clean hydrogen and to treat environmental pollution, which is considered to be a preferred method in order to solve environmental and energy issues [12]. Therefore, the formation of heterostructures is proposed as they can combine the benefits of each material and improve the separation rate of photoexcited carriers [13]. It is also indicated that the contact surface area of the heterojunction interface provides significant enhancement to the photoresponse capability in the route of the photocatalytic redox reaction [14].

Transition metal tungsten disulfides and tungsten diselenides have been broadly researched as excellent semiconductor photocatalysts [15,16,17]. Because of their outstanding photoelectric capability and particular band gap properties, they possess better potential in the field of photocatalytic activity. As one of the favorable photocatalysts, WS_2_ and WSe_2_ have been studied for photocatalytic hydrogen evolution owing to their appropriate band gaps and proper energy band gap positions. Then, the inactive activities of photoinduced charge carriers separate and confine the photocatalytic hydrogen energy performance of the bare WS_2_ and WSe_2_ components [18]. On the other hand, the rapid recombination of photoexcited charge carriers of the bare WS_2_ and WSe_2_ seriously suppress its efficient utilization in the environmental field. In this heterostructure, two necessities are essential in order to quickly separate the electron and hole pairs from the reduced material and the oxidized material.

Based on the above studies, the formation of WSSe nanocomposites could be favorable to the powerful separation and transfer of photoexcited charge carriers and enhance their photocatalytic efficiency. The photocatalytic activity of WSSe nanocomposites was systematically studied through photocatalytic hydrogen evolution and MB removal under visible light irradiation. Our research study will open a new route for designing novel nanocomposites with outstanding functional capabilities for future industrial applications.

## 2. Materials and Methods

### 2.1. Materials

Sodium tungstate dihydrate (Na_2_WO_4_·2H2O, >99%), L-cysteine (C_3_H_7_NO_2_S, >97%), selenium powder (Se, >99.99%), hydrochloric acid (HCl, >37%), Methylene blue (MB), and ethanol (C_2_H_5_OH, >99.7%) were obtained from Sigma-Aldrich. Deionized (DI) water (resistivity > 18.0 MΩ) was used as an experimental water source in further experimental procedures. All of the reagents were applied without any further purification.

### 2.2. Synthesis

The water-soluble WSSe nanocomposite photocatalysts were fabricated using a facile hydrothermal process. Firstly, 65 mg of Na_2_WO_4_·2H_2_O was added to 15 mL of DI water and stirred for 10 min. Once completed, 0.1 M HCl was used in order to adjust the pH to 6.8. Next, 25 mg of L-cysteine, selenium powder (1 wt%, 5 wt% and 10 wt%), and 50 mL of water were added to the mixture solution and stirred for 15 min. The resulting mixture was transferred to a Teflon-lined autoclave and kept at 180℃ for 24 h in an oven. Once the reaction was completed, the precipitates were centrifuged for 30 min at 12,000 rpm. The resulting product was washed several times with alcohol, deionized water, and acetone. The washed precipitates were dried at 80℃ in a vacuum drying oven (overnight). Eventually, a series of WSSe nanocomposites with a diverse Se mass ratio of 1 wt% (0.9 mg), 5 wt% (4.5 mg), and 10 wt% (9 mg) were arranged, and served as WSSe-1, WSSe-2, and WSSe-3, respectively.

### 2.3. Characterization

The field emission scanning electron microscopy (SEM) and transmission electron microscopy (TEM) were carried out in order to observe the microstructure and morphology of the samples. The chemical composition of the samples was carried out by X-ray photoelectron spectroscopy (Thermo Fisher Scientific, Waltham, MA, USA) and was also recorded. Brunauer–Emmett–Teller (BET) were characterized in order to measure the pore volume, surface area, and pore distribution. The crystal phase and structure of the as-fabricated samples were utilized with a powder X-ray diffractometer (Bruker D8 Advance, USA) applying Cu Ka radiation (k = 0.154168 nm). The optical spectra were recorded by a UV–Vis reflectance spectrophotometer (Hitachi U-4100, Japan.) and photoluminescence (PL). The photochemical property was determined using an electrochemical workstation with a 0.2-M Na_2_SO_4_ solution as the electrolyte. Electron spin resonance (ESR) signals of the radicals captured were performed with visible light activity on a Bruker A300 spectrometer.

### 2.4. Photocatalytic Hydrogen Production

The hydrogen production test was analyzed by quantitatively and periodically extracting gas from a reactor with visible light irradiation. Driving nitrogen in to replace the air in the reactor was sufficient in order to assure the atmospheric pressure in the reactor. In this study, 10 mg of the photocatalyst was used in a mixture solution (50 mL) including 30 mmol Na_2_S and 20 mmol Na_2_SO_3_ as sacrificial agents. The mixture solution was evacuated in order to extract air entirely before the activities. The pumping light source adopted was a 300 W Xe lamp with λ > 420 nm filter. The mixture solution was stirred repeatedly under the reaction. The amount of hydrogen production during the light activity was determined using gas chromatography (GC-7890, TCD) coupled with a thermal conductivity detector in order to measure the evolution of H_2_.

### 2.5. Photocatalytic MB Degradation Activity

The photocatalytic capability of the prepared nanocomposite samples was used by removing the efficiency of an organic compound (MB) in an aqueous solution by applying 300 W Xe lamp as a light source. Before the photoreactions, 10 mg of the as-prepared sample was dispersed in the 100 mL (10 ppm) MB solution and the reaction solution was continuously stirred for 30 min under darkness in order to achieve the adsorption–desorption balance between the organic compound and photocatalyst’s surface. Accordingly, the reaction solution was irradiated with a Xe lamp source, and then a 5 mL suspension was taken out in the same conditions. Once the photocatalyst was removed through centrifugation, the concentration of MB was determined using the UV–Vis spectrophotometer. Moreover, the trapping tests were further performed in order to analyze the photoexcited active species by conducting a particular amount of scavengers EDTA-2Na, 1,4-benzoquinone (BQ), and 2-propanol (IPA) into the reaction system.

## 3. Results

### 3.1. XRD and UV-Vis Analysis

Figure 1 displays the XRD patterns of the WS_2_, WSe_2_, WSSe-1, WSSe-2, and WSSe-3. samples recorded by combining various amounts of Se. As exhibited in the results, with the addition of the Se, the diffraction peak position of the WSSe does not obviously change, and no other impurity peaks appear. As the counting of Se is increased from 10 to 30 mg, the XRD analysis of WSSe-X (X = 1, 2, and 3) becomes increasingly shifted within the range of 13.5° to 14.3°, corresponding to (002) planes of WS_2_ and WSe_2_ (JCPDS 84-1398 and 96-901-2194) [15,16,17]. The major characteristic peaks of the (002) plane can be observed in the patterns of WSSe-1, WSSe-2, and WSSe-3. With the increase of the Se content, the position of the characteristic peaks of the (002) plane in the WSSe-1, WSSe-2, and WSSe-3 patterns gradually move toward the low 2 theta degree, which proves that the WSSe-1, WSSe-2, and WSSe-3 heterojunctions have been successfully constructed. This result also indicates the successful chemical combining of Se and WS_2_ in the WSSe composite. The results demonstrate that the facile hydrothermal process can achieve a heterostructure without change the lattice structure. UV–Vis was conducted in order to analyze the optical properties of the samples. Figure 1b displays that the absorption edge of WSSe-X (WSSe-1, WSSe-2, and WSSe-3) is about 450~520 nm, indicating that the WSSe-X can react well to visible light range. The absorption edge of the WSSe-3 heterojunction has a redshift compared with WSSe-1 due to the interaction between WS_2_ and WSe_2._ It means that the generated WSSe-3 nanocomposite presented the enhanced absorption ability of visible light, which was favorable in order to generate more photoexcited electrons and holes for beginning the redox reactions. The band gap (Eg) energies of WSSe-X were calculated using the Tauc equation ((αhυ)^2^ = A (hυ − Eg)^n^) [19]. Figure 1b inset reveals that the Eg of WSSe-1, WSSe-2, and WSSe-3 are 2.74, 2.66, and 2.56 eV, respectively.

### 3.2. SEM and TEM Analysis

The morphology and microstructure of the WS_2_, WSe_2_, and WSSe-3 samples were also measured by SEM (Figure 2) and TEM (Figure 3). Figure 2a,b shows the SEM image of the pure WS_2_ and WSe_2_ samples. It is obvious that the bulk of WS_2_ and WSe_2_ materials that are severely agglomerated were in the range of an average width 10–20 μm. As shown in Figure 2c, the WSSe-3 has a particle size of nearly 2–3 μm, and a sphere-like morphology on its surface developed by the rough stacking of particles. The rough organization of particles contributes not only to the volume of space but also endows support for the firm attaching of the WS_2_ and WSe_2_ samples. Noticeably, compared with WS_2_ and WSe_2_, the surface of WSSe reveals that it is obviously loose, which will be profitable for WSSe-3 in order to serve as a better specific surface area and to be easier to combine with other materials. The TEM images of the WSSe-3 composites were further provided in Figure 3a–f. It was observed that the WS_2_ was combined with a WSe_2_ and with an evident interface between the two materials, implying the presence of a heterojunction structure. In addition, Figure 3b shows the HRTEM image of the WSSe-3 samples, and the interlayer distances of 0.28 and 0.34 nm were corresponding to the (002) lattice plane of WS_2_ and (110) lattice planes of WSe_2_, respectively [15,16]. The elements of W, S, and Se were smoothly dispersed on the entire composite in the EDS mapping (Figure 3c–f), suggesting the presence of WS_2_ and WSe_2_ in nanocomposites. Moreover, it was noted that the areas of the elemental distribution of WS_2_ with a sphere shape were smaller than that of the WSe_2_ sample. It also suggested that the WSe_2_ nanoparticles had adhered to the surface of WS_2_, thereby confirming that the heterojunction with WSSe was well fabricated. This heterostructure would mainly expand the contact surface area between WSe_2_ and WS_2_, thus improving the absorption ability of small molecules (organic pollutants and water molecules). The results reveal that improving the photocatalytic activity of the composite due to the acceleration of the separation of the charge carriers [20]. Therefore, we confirmed the successful formation of the WSSe-3 nanocomposite.

### 3.3. XPS and N_2_ Adsorption–Desorption Analysis

X-ray photoelectron spectroscopy (XPS) was performed in order to check the chemical composition and surface element status in the as-prepared WSSe-3 sample. Figure 4 displays the high-resolution XPS spectra of different elements for the WSSe-3 sample. As displayed in Figure 4a, the feature peaks of W could be divided into three peaks, among which the feature peaks at 32.1, 34.2, and 37.6 eV and refer to the W 4f 7/2, W 4f 5/2, and W 5p 3/2, respectively. In Figure 4b, a high-resolution spectrum of the S 2p spectrum can be divided into S 2p3/2 and S 2p1/2 spin–orbit doublets at 162.8 and 161.7 eV, implying that S^2-^ existed in the WSSe-3. Then, two splitting feature peaks of the Se 3d spectrum in WSSe-3 examined at 53.8 eV and 54.6 eV can be referred to as Se 3d5/2 and S 3d3/2 (Figure 4c), determining that the chemical status of the Se element was Se^3+^, relative to the previous report [21,22]. For the above XPS test results of the as-fabricated photocatalyst, it showed that the nanocomposite was well prepared by a facile hydrothermal process and there was a vigorous interaction between WS_2_ and WSe_2_, which was in agreement with the XRD results. In order to check the BET-specific surface area of the nanocomposites, N_2_ adsorption–desorption tests were performed (Table 1). The specific surface areas of WS_2_, WSe_2_, and WSSe-3 are 33.6, 23.3, and 67.4 m^2^ g^−1^, respectively, demonstrating that the specific surface area of WSSe-3 increases significantly after forming with a nanocomposite. The nanocomposites have a large specific surface area and have a wide range of light absorption (i.e., from the ultraviolet to the visible region) for a strong photocatalytic efficiency [23]. Furthermore, the addition of selenium can adjust the surface electronic structure, change the surface morphology of the catalyst, and then affect the catalytic activity [24]. The high surface area of the WSSe nanocomposites increases ion adsorption, resulting in increased ionic conductivity and specific capacitance of the material, in good line with the previous report [25]. Moreover, compared with WS_2_ and WSe_2_, the average pore diameter (8.7 nm) of WSSe-3 is improved. Although the pore diameter is reduced, the overall photocatalytic performance of the fabricated nanocomposite is better for the corresponding bare components due to the reduction of the fabricated nanocomposite band gap energy [26]. The construction of the photocatalyst interface contributes to the expanded life of the charge carriers and practical electrons and holes separation, which in turn perform at a higher photocatalytic level. The pore volume displays a similar trend as that of the specific surface area, with the pore size that is smaller in the nanocomposites where Se was substituted. The specific surface area of the regular samples is typically associated to the particle size, with smaller particles possessing a greater surface area [27]. These results suggest that WSSe nanocomposites with the excellent surface area and pore size distribution centering at 67.4 m^2^/g and 8.7 nm could better improve the photocatalytic capability than that with bare WSe_2_ and WS_2_. According to the results above, when fabricating WSSe nanocomposites for a better potential hydrogen production and MB removal, both their high specific surface area and suitable pore size distribution should be considered.

### 3.4. Photocatalytic Hydrogen Production Studies

As displayed in Figure 5a,b, the hydrogen production rate of WSSe-3 enhanced by 12 and 11 times that of WS_2_ and WSe_2_, respectively. The total production of H_2_ followed the order: WSSe-3 (3647.4 μmol·g^−1^) > WSSe-2 (3145.6 μmol·g^−1^) > WSSe-1 (2547.3 μmol·g^−1^) > WSe_2_ (324.5 μmol·g^−1^) > WS_2_ (306.2 μmol·g^−1^). When the addition of the photocatalyst was excessive, owing to the WSSe-3 nanocomposites aggregation, this suggested a reduction in the surface area of WSSe-3 and thus hindered the transfer ability of the charge carriers [28,29]. However, all WSSe composites show an improved photocatalytic hydrogen evolution efficiency, thereby implying that the electron localization and boost interaction at the WSSe-3 sample interface capably accelerate the charge transfer and separation ability. Moreover, excessive nanocomposites may occur in the recombination of electron and hole pairs, hence decreasing the photocatalytic efficiency. By contrast, a slight amount of the photocatalyst will cause the interface between WS_2_ and WSe_2_ to become too restricted to separate and transport the photoexcited electrons and holes pairs, which cannot mainly suppress the recombination rate of the photoexcited charge carriers. As exhibited in Figure 5c, with the addition of the WSSe-3 nanocomposites, the amount of hydrogen production increases gradually until the amount of WSSe-3 nanocomposites approaches 20 mg. Moreover, the hydrogen production rate is gradually suppressed with the increasing WSSe-3 nanocomposites. This reveals that the water molecules in the solution could be adsorbed by the photocatalysts and mainly employed when the amount of the WSSe-3 nanocomposites is suitable. When the mass of WSSe-3 nanocomposites is over 20 mg, the excess photocatalyst stock in the mixture solution inhibits the efficient charge carrier’s transport and photon adsorption of the nanocomposite and hence decrease the amount of photons on the sample surface, suggesting an decreased hydrogen evolution efficiency. In order to investigate the reusability and the chemical stability of the WSSe-3 nanocomposite in the hydrogen evolution rate, a cycling test was carried out in the same conditions. As displayed in Figure 5d, the amount of hydrogen generated in the fourth cycle was obviously decreased compared with the original case. The hydrogen production rate remained at nearly 87.4% compared with the original one. When 20 mg of WSSe-3 nanocomposites were increased again, the amount of the hydrogen production rate evidently increased, implying that the photocatalytic of the WSSe-3 nanocomposites was the primary factor for the increase in the hydrogen production. At the end of the cycling performance analysis, the XRD analysis of the sample was conducted again. As displayed in Figure 5e, the crystal structure and morphology of WSSe-3 do not show an obvious change before or after the cycling test, which further demonstrates the excellent chemical stability and repeatability of the nanocomposite in the photocatalytic hydrogen evolution efficiency. Therefore, the hydrogen evolution rate of the WS_2_ mixtures is 306.2 μmol g^−1^ h^−1^, which is significantly lower than the WSSe-X heterojunctions. The above results implied that the as-prepared WSSe-3 nanocomposite can be applied as a potential hydrogen generating photocatalyst. The performance differences of the tungsten based photocatalysts for the hydrogen evolution reaction is shown in Table 2 [30,31,32,33].

### 3.5. Photocatalytic MB Degradation Studies

It can be noted from Figure 6a that the feature absorption intensity of MB at 664 nm apparently decreased with an increasing irradiation time, confirming that MB was practically degraded by the WSSe-3 nanocomposite. The above photocatalytic results revealed that the preparation of the WSSe-3 nanocomposite diminished the accelerated separation rate and the space charge transfer distance of the photoexcited charge carriers, thus improving the photodegradation performance. In order to further check the photodegradation performance of the photocatalyst, the photodegradation of the MB pollutant was also conducted. Figure 6b displays that the adsorption equilibrium test in the dark for 20 min. The adsorption amount of MB via all photocatalysts is negligible. The blank test shows that the self-photocatalysis of MB could be ignored, indicating that the property of MB is stable in the photocatalytic activity. WS_2_ and WSe_2_ have a poor capability for the MB removal with visible light irradiation, which are 34.8% and 49.2% within 60 min, respectively. Noticeably, WSSe-1, WSSe-2, and WSSe-3 approach a better photocatalytic capability for the MB removal, which is 79.3%, 86.4%, and 88.5%. Corresponding to the photocatalytic hydrogen evolution, the operation series of the photocatalytic MB removal is as follows: WSSe-3 > WSSe-2 > WSSe-1 > WSe_2_ > WS_2_. Moreover, the MB degradation performance of WSSe-3 is 2.6 and 1.8 times that of WS_2_ and WSe_2_ samples, respectively. Figure 6c displays the cycling experiment of WSSe-3 for the photo degradation of the MB pollutant. The photodegradation efficiency decreases slightly from 88.6% to 83.2% after six repeated cycling tests. Observing the cycling experiment of the WSSe-3 photocatalytic process, the MB degradation rate slightly decreases due to less photocorrosion caused by the remaining charges on the WSSe-3 surface. These results suggested that the fabricated WSSe-3 nanocomposite could be adopted for an excellent stability and repeatability during the long-term photodecomposition system. Such outstanding photocatalytic efficiency of the WSSe nanocomposite is comparable or even better than those nanocomposites for the photodegradation of the MB pollutant reported in the literature (as summarized in Table 3) [27,34,35,36,37].

To further confirm the active species of the photocatalyst, the influences of various capture agents during the photocatalytic activity were proved. The capture agents are isopropanol (IPA) for ·OH, benzoquinone (BQ) for ·O_2_^−^, and EDTA-2Na for h^+^. When IPA is added to the photocatalytic activity, the photocatalytic reaction slightly diminishes, implying that ·OH has a weak effect on the MB removal in Figure 6d. On the other hand, when BQ or EDTA-2Na is added to this test, the photocatalytic efficiency visibly decreases, implying that h^+^ and ·O_2_^−^ are the major active species [38,39]. Furthermore, the ESR analysis (Figure 6e) indicates that WSSe-3 has the DMPO-·O_2_^−^ adduct signal, which also confirms that ·O_2_^−^ exists in the photocatalytic activity. Electrochemical impedance spectroscopy (EIS) was performed in order to check the transportation and charge carrier’s separation rate of the nanocomposite surface [40]. In Figure 6f, the Nyquist semicircle radius of WSSe-3 is significantly smaller than that of the pure WS_2_ and WSe_2_, confirming that the transfer resistance of the photoexcited electrons and holes in WSSe-3 is the smallest among these catalysts. It was indicated that the WSSe-3 improved the charge transfer ability for the composite. Firstly, we examined the linear sweeping voltammetry (LSV) curves of WS_2_, WSe_2_, WSSe-1, WSSe-2, and WSSe-3 in a 0.1 M KOH solution at a scan rate of 2 mV s^−^^1^ (Figure 6g). Obviously, the WSSe-3 nanocomposite displays a better LSV performance than the other photocatalysts. These results further confirm that the lattice defects on the WSSe-3 nanocomposite may be more useful for supplying more photocatalytic reaction sites, mainly by enhancing the redox activity area and further strengthening the MB degradation and hydrogen evolution action performance. In order to further study the charge carrier’s transfer and separation capability of the nanocomposites, a PL measurement was conducted. Figure 6h exhibits the PL spectra of WS_2_, WSe_2_, and WSSe-3 under 420 nm excitation. Compared with WS_2_ and WSe_2_, the PL peak intensity of WSSe-3 decreased significantly. The PL intensity decreased in the order of WSSe-3 < WSe_2_< WS_2_, clearly describing that the WSSe-3 sample presented a better electrons and holes separation capability, which was in agreement with the above EIS analysis results. The results demonstrated that the WSSe-3 nanocomposite can mainly transfer photoexcited electrons and holes in order to prohibit their recombination ability. All of the above results confirm that the photoexcited electrons and holes in WSSe-3 could be effectively transferred and separated.

### 3.6. Proposed Photocatalytic H2 and MB Degradation Mechanism

The probable photoexcited charge carrier’s transport paths for the enhanced photocatalytic efficiency was recommended in Figure 7. Both WS_2_ and WSe_2_ were induced and excited charge carriers on their conduction bands and valence bands under visible light irradiation conditions due to their narrow band energies. This result of the photocatalytic performance towards the H_2_ evolution and the MB degradation under visible light illumination was similar to the previous studies [27,30,31,32,33,34,35,36,37]. Accordingly, the photo-induced charge carriers were transferred from CB and VB of WS_2_ to the consistent positions of WSe_2_ due to the band gap energy difference. Because of the nanocomposites, the confined charge carriers withhigh energy enhanced in the heterostructure can be transported rapidly to the photocatalyst surface through the tunneling influence in order to contribute to the chemical redox activity. Therefore, some section of the photoexcited electrons and holes might also directly contribute to the redox reduction owing to the nanocomposites developed in the heterojunction of WSSe-3 nanomaterials. The photoexcited charges collected on the surface of the photocatalyst can directly contribute to the reduction of H_2_O to H_2_ under the hydrogen production process. Then, the nanocomposite is the main active site for the reaction. For the photodegradation procedure of MB, the electrons on the surface of the photocatalyst can respond with O_2_ to generate ·O_2_^−^ and H^+^ would be reduced to H_2_, which is the primary active species for the degradation of the MB reaction and hydrogen production. At the same time, the holes directly oxidized on the MB organic molecule with H_2_O in order to generate ·OH. This is in agreement with the ESR trapping results. The possible mechanism for the enhanced photocatalytic activity from the above experiment analysis were as follows: (1) the extended specific surface area was profitable for creating more active sites and a strong photoactivity. (2) the improved visible light absorption range could increase the light capture efficiency in order to enhance the photocatalytic utilization. (3) the synthesized heterostructure with the preferred interface contact offers a better transfer pathway for the charge carriers, which can promote photoexcited electrons and holes separation capability in order to enhance the photocatalytic efficiency.

## 4. Conclusions

In conclusion, the novel WSSe nanocomposites were synthesized by the hydrothermal process, which led to the formation of the intimated interfacial contact and heterojunction for energy conversion and environmental remediation. This study demonstrated that the as-fabricated WSSe-3 was beneficial to the migration of photoexcited charge carriers, and could further enhance its photocatalytic hydrogen production activity and the MB degradation under visible light irradiation. In addition, the photocatalytic activity of WSSe-3 rose to 3647.4 μmol g^−1^ h^−1^ (the H_2_ production rate) and 88% (photocatalytic efficiency of the MB degradation), which were 12 and 2.5 times higher than that of the bare WS_2_, respectively. The outstanding improvement of the photocatalytic system might be due to the nanocomposite structure and the heterojunction effect between WS_2_ and WSe_2_. Furthermore, the WSSe-3 nanocomposite exhibited better reusability and chemical stability. This study not only improves the light absorption efficiency of the prepared material, but also produces more photocatalytic active sites for shortening the electron’s transport path. This study provides valuable reference to heterojunction photocatalysts in the water splitting field and for further exploration of the photocatalytic mechanism for potential sustainable energy production in the future.

## Figures and Tables

**Figure 1 materials-15-05616-f001:**
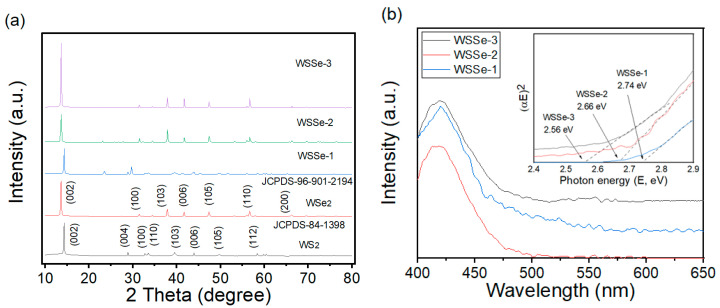
(**a**) XRD patterns of the as-prepared samples; (**b**) UV–vis spectra of WSSe-1, WSSe-2, and WSSe-3 samples; plots of (αhυ)^2^ versus hυ for the bandgap energies of the photocatalysts in the inset.

**Figure 2 materials-15-05616-f002:**
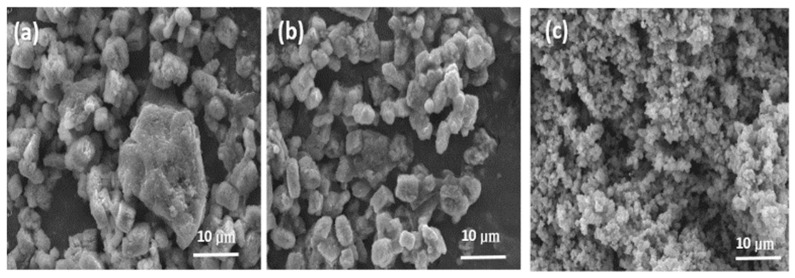
SEM images of (**a**) WS_2_, (**b**) WSe_2_, and (**c**) WSSe-3.

**Figure 3 materials-15-05616-f003:**
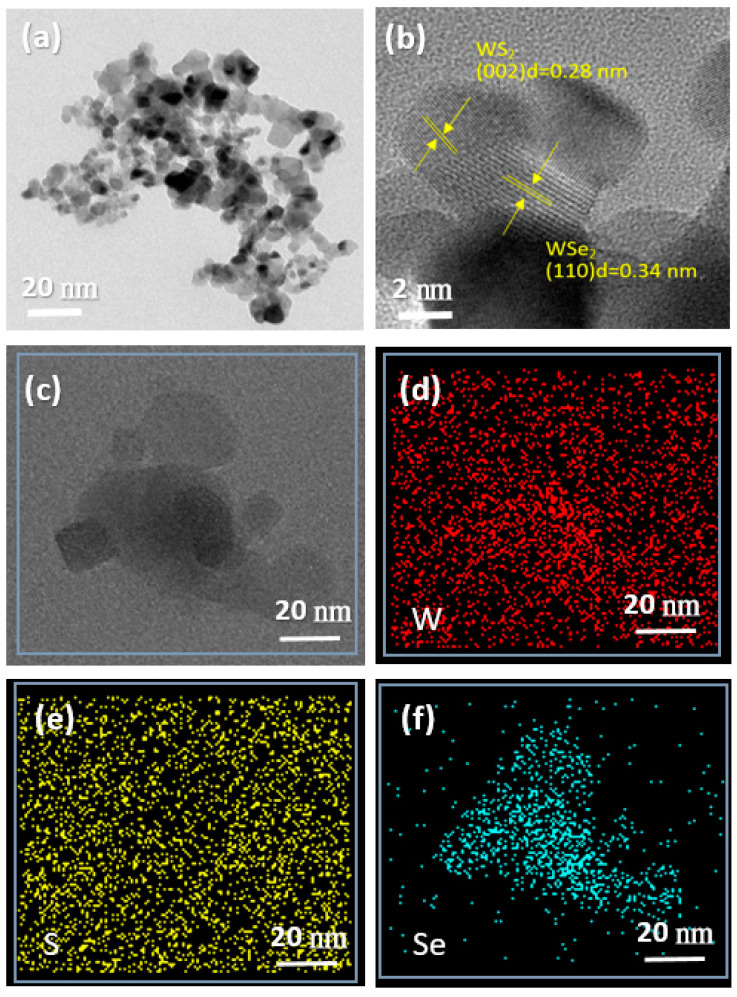
TEM images of (**a**) WSSe-3; HRTEM images (**b**) and element mapping images (**c**–**f**) of WSSe-3.

**Figure 4 materials-15-05616-f004:**
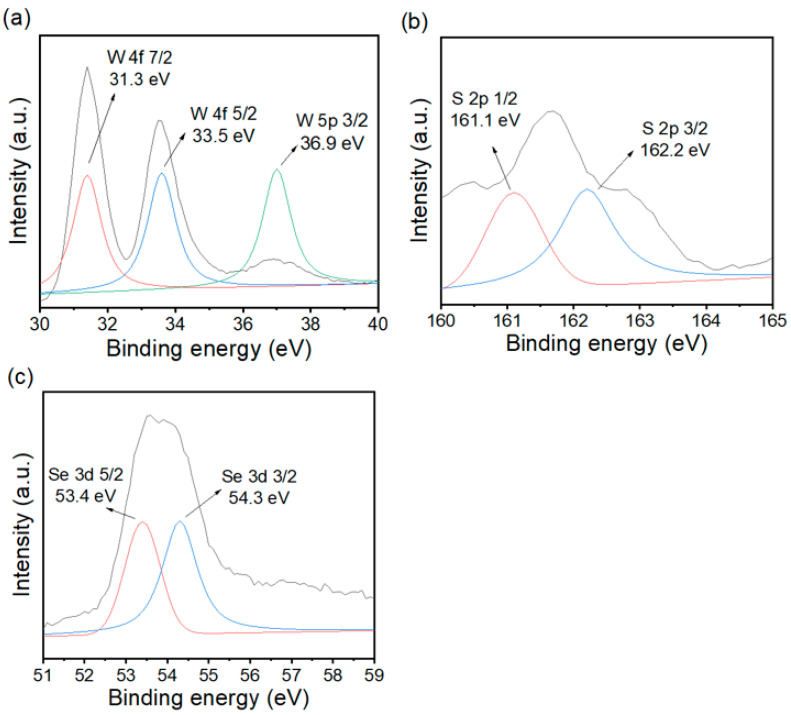
XPS survey spectra of WSSe-3. High-resolution XPS spectra of the (**a**) W 4f, (**b**) S 2p, and (**c**) Se 3d.

**Figure 5 materials-15-05616-f005:**
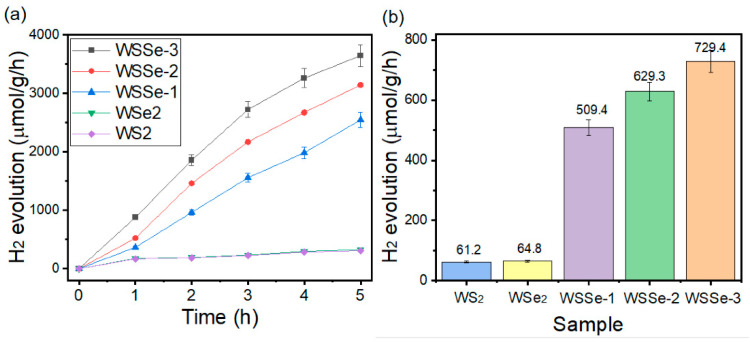
H_2_ evolution of (**a**) WS_2_, WSe_2_, and WSSe-3 and H_2_ production comparison of (**b**) WSSe-1, WSSe-2 and WSSe-3. (**c**) H_2_ production with various usage amounts of WSSe-3. (**d**) H_2_ production of the cycling performance, (**e**) XRD analysis of WSSe-3 before and after the hydrogen production activity. (The error bars represent results of three duplicates.).

**Figure 6 materials-15-05616-f006:**
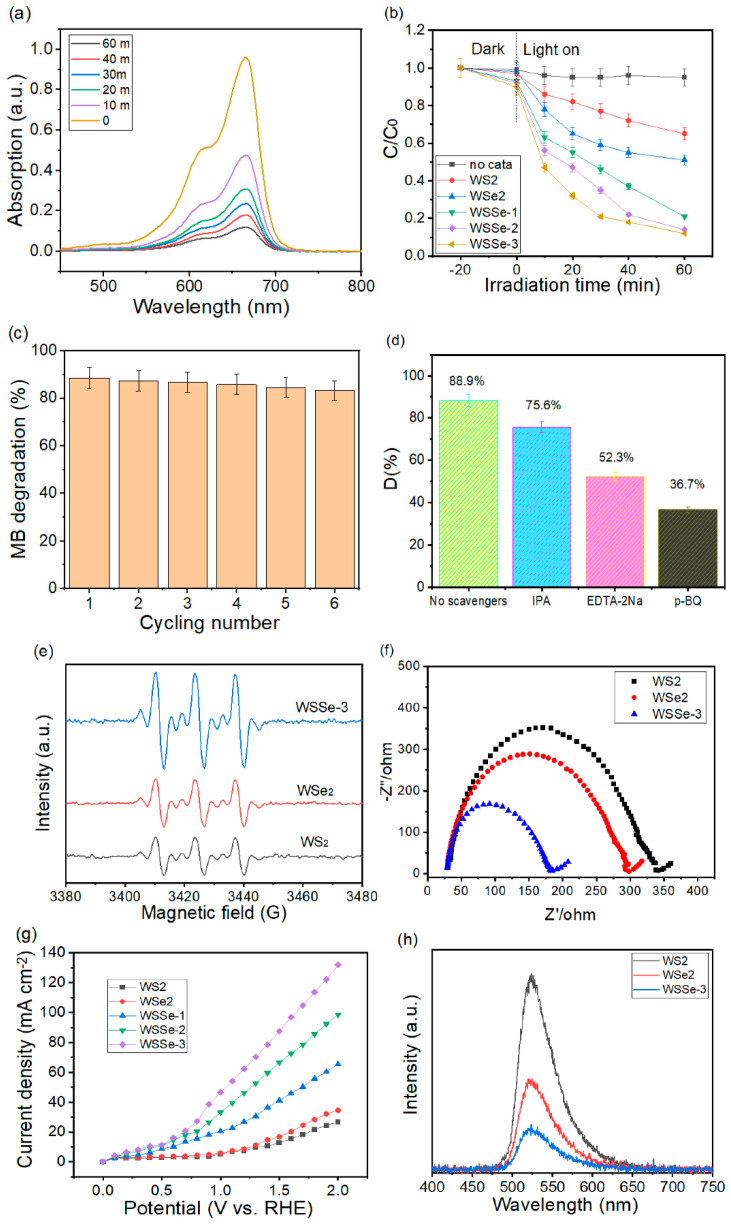
(**a**) MB spectra adsorption changes during the photocatalytic process, (**b**) photo degradation curves of MB, (**c**) cycling test for the WSSe-3 sample, (**d**) reactive species trapping tests, and (**e**) DMPO spin-trapping ESR spectra of DMPO-·O_2_^−^. (three repeat tests in order to estimate error bars), (**f**) electrochemical impedance spectroscopy of the as-fabricated photocatalysts, (**g**) LSV curves, and (**h**) photoluminescence spectroscopy (PL) of WS_2_, WSe_2_, and WSSe-3. (The error bars represent results of three duplicates.).

**Figure 7 materials-15-05616-f007:**
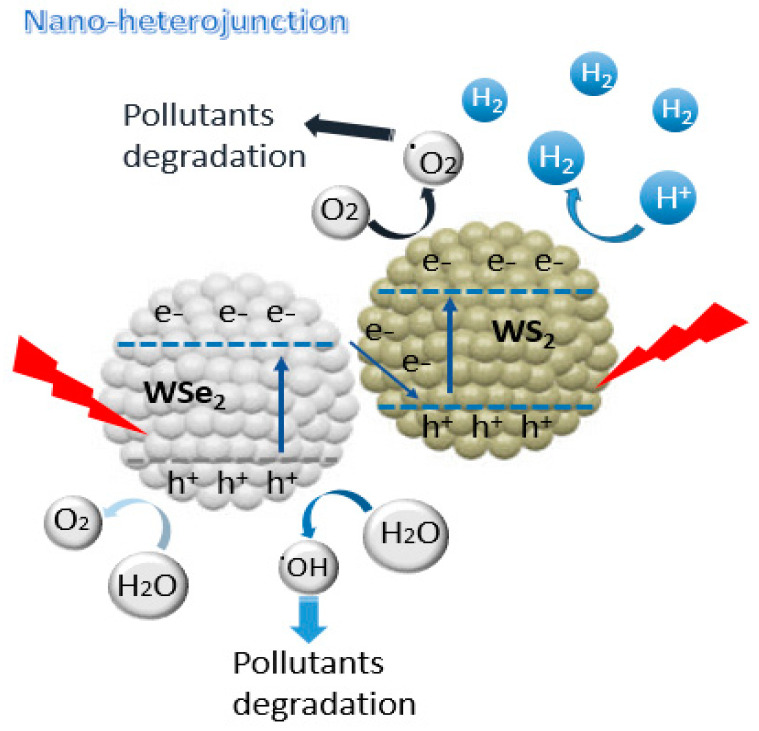
Schematic diagram of the proposed heterojunction photoelectron mechanism of the WSSe photocatalytic system under visible light illumination.

**Table 1 materials-15-05616-t001:** The physical adsorption parameters of the photocatalysts of WS_2_, WSe_2_, and WSSe-3 were measured three times.

Samples	Specific Surface Area (m^2^/g, BET) ^a^	Total Pore Volume (cm^3^/g, BET) ^b^	Average Pore Diameter (nm, BJH) ^b^
**WSe_2_**	33.6 ± 5	0.182 ± 0.02	38.4 ± 5
**WS_2_**	23.3 ± 5	0.113 ± 0.02	48.5 ± 5
**WSSe-3**	67.4 ± 5	0.325 ± 0.02	8.7 ± 3

^a^ Received from BET analysis. ^b^ Relative pressure (P/P_0_) was 0.99.

**Table 2 materials-15-05616-t002:** Studies of the tungsten-based photocatalysts for the hydrogen production.

Photocatalyst	Tafel Slope (mV/dec)	Overpotential at 10 mA/cm^2^	Exchange Current Density (mA/cm^2^)	Electrolytes	Reference
Co_0.5_Fe_0.5_WO_4_	36.3	360 mV	10	1.0 M KOH	[30]
Fe-WOx	51.7	0.38 V	10	1.0 M KOH	[31]
Fe_2_P-WO_2.92_/NF	46.3	267 mV	100	1.0 M KOH	[32]
FeNiW-LDH	55.7	202 mV	10	1.0 M KOH	[33]
WSSe	65.2	252 mV	10	0.1 M KOH	This work

**Table 3 materials-15-05616-t003:** Studies of nanocomposite photocatalysts for the degradation of the MB pollutant.

Photocatalyst	Catalyst Loading	Methylene Blue	Time	Degradation Efficiency	Reference
rGO/AgVO_3_	20 mg	25 ppm	2.5 h	88%	[27]
Bi_2_Te_3_/CdS/CuFe_2_O_4_	50 mg	40 ppm	2 h	97%	[34]
CuO–Cu_2_O	40 mg	5 ppm	4 h	90%	[35]
ZnO	30 mg	20 ppm	1.5 h	98%	[36]
R–TiO_2_	50 mg	10 ppm	6 h	90%	[37]
WSSe	20 mg	10 ppm	1 h	89%	This work

## Data Availability

Not applicable.

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
