# Peer review of "WSSe Nanocomposites for Enhanced Photocatalytic Hydrogen Evolution and Methylene Blue Removal under Visible-Light Irradiation"

_materials, 2022, doi:10.3390/ma15165616_

Round 1

Reviewer 1 Report

The manuscript is well organized and has high potential to contribute to the development of an efficient photocatalyst. The is sufficient to publish in the target journal and does not need to add more. Few minor comments to the authors

> I have doubts about the pore size calculation. Particle size and pore size do not make any correlation. 

>Please record TEM images and conform pore size.

>Figures quality need to improve. 

>Please discuss how surface area and pore size in terms of diffusion of reactant playing important role in overall catalyst performance. 

> Authors claimed that this catalyst is the best, please compare your results with previously published data. 

Author Response

Journal: Materials (ISSN 1996-1944)

Manuscript ID: materials-1860580

Type: Article

Title: WSSe nanocomposite for enhanced photocatalytic hydrogen evolution and Methylene blue removal under visible-light irradiation

Author's Reply to the Review Report (Reviewer 1)

Comments and Suggestions for Authors

The manuscript is well organized and has high potential to contribute to the development of an efficient photocatalyst. The is sufficient to publish in the target journal and does not need to add more. Few minor comments to the authors

> I have doubts about the pore size calculation. Particle size and pore size do not make any correlation. 

Reply: Thank you for your valuable suggestion. It’s true that our original statement is not clearly enough. According to the prior study, the lowering of the surface area of the nanocomposite can be attributed to the increase in grain size and particle growth during synthesis. The mean pore size of pure WSe2 and WS2 are estimated at 38.4 nm and 48.5 nm, however, it decreases to 8.7 nm after forming WSSe nanomaterials. Moreover, specific surface area of the pure WSe2 and WS2 are evaluated as 33.6 m2/g and 23.3 m2/g, however it increases to 67.4 m2/g after integrating with WSSe nanomaterials (Table 1). For the smaller particle size, WSSe nanomaterials had a higher active surface area than that of pure sample. The high surface area of the WSSe nanomaterials increases ion adsorption, resulting in increased ionic conductivity and specific capacitance of the material, in good line with previous report. 25,26 At the same time, the nano-scale WSSe photocatalyst can remarkably suppress the photogenerated electron-hole pairs re-combination procedures, subsequently, support in refining the photocatalytic efficiency of the nano-scale WSSe photocatalyst. Although the pore diameter is reduced, the overall efficiency of the synthesized nanocomposite is superior to the corresponding bulk components due to the reducing synthesized nanocomposite band gap, and it also prevents the accumulation of the nano-scale WSSe photocatalyst. It has been described in Page 5, Lines 225~231: “ The high surface area of the WSSe nanocomposites increases ion adsorption, resulting in increased ionic conductivity and specific capacitance of the material, in good line with previous report.25 Moreover, compared with WSSe-3 (0.325 cm3 g-1), the average pore diameter (8.7 nm) of WSSe-3 is improved. Although the pore diameter is reduced, the overall photocatalytic performance of the fabricated nanocomposite is better to the corresponding bare components due to the reducing fabricated nanocomposite band gap energy.26

  1. Pataniya, P. M.; Sumesh, C.K. Enhanced electrocatalytic hydrogen evolution reaction by injection of photogenerated electrons in Ag/WS2 nanohybrids, Surf. Sci. 2021, 563, 150323.
  2. Keerthana, B. G. T.; Murugakoothan, P. Synthesis and characterization of CdS/TiO2 nanocomposite: Methylene blue adsorption and enhanced photocatalytic activities, Vacuum 2019, 159, 476.

>Please record TEM images and conform pore size.

Reply: Thank you very much for pointing out this. According to your suggestion, we have revised the TEM images in the revised manuscript.

>Figures quality need to improve. 

Reply: Thank you for your valuable suggestion. All images have been revised as follow above.

>Please discuss how surface area and pore size in terms of diffusion of reactant playing important role in overall catalyst performance. 

Reply: It’s true that our original statement is not clearly enough. According to your suggestion, we have added some additional description and cited some references in the revised manuscript. It has been described in Page 5-6, Lines 231~242: “ The construction of the photocatalyst interface contributes to the expanded life of the charge carriers and practical electrons-holes separation, which in turn conduct to a higher photocatalytic performance. The pore volume displays a similar trend as that of the specific surface area, whatever, the pore size is smaller in the nanocomposites where Se was substituted. The specific surface area of regular samples is typically associated to the particle size, with smaller particles possessing a huger surface area.27 These results suggest WSSe nanocomposites with the excellent surface area and pore size distribution centering at 67.4 m2/g and 8.7 nm could better improve the photocatalytic capability than that with bare WSe2 and WS2. According to the results above, when fabricating WSSe nanocomposites for better potential hydrogen production and MB removal, both of high specific surface area and suitable pore size distribution of them should be essentially considered. ”

  1. Mansha, M. S.; Iqbal, T. Experimental and theoretical study of novel rGO/AgVO3 nano-hetrostructures for their application as efficient photocatalyst, Opt. Mater. 2022, 131, 112591.

> Authors claimed that this catalyst is the best, please compare your results with previously published data. 

Reply: Thanks for your suggestions. Photocatalytic process is an emerging, efficient and energy-save technology for the removal of organic contaminants from the water environment. With the development of functional photocatalysts, nanocomposites have become one of the most popular emerging co-catalysts due to its high photocatalytic activity, strong adsorptivity, low cost and non-toxicity, especially applied to the photocatalytic degradation of organic contaminants. It has been described in Page 10, Lines 446~449: “ Such outstanding photocatalytic efficiency of WSSe nanocomposite is comparable or even better than those nanocomposites for photodegradation of MB pollutant reported in the literature (as summarized in Table 2).”

Table 2. Studies of nanocomposite photocatalysts. for the degradation of MB pollutant.

Photocatalyst

Catalyst loading

Methylene blue

Time

Degradation

efficiency

reference

rGO/AgVO3

20 mg

25ppm

2.5h

88%

27

Bi2Te3/CdS/CuFe2O4

50 mg

40ppm

2h

97%

34

CuO–Cu2O

40 mg

5ppm

4h

90%

35

ZnO

30 mg

20ppm

1.5h

98%

36

R–TiO2

50 mg

10 ppm

6h

90%

37

WSSe

20 mg

10ppm

1h

89%

This work

  1. Mansha, M. S.; Iqbal, T. Experimental and theoretical study of novel rGO/AgVO3 nano-hetrostructures for their application as efficient photocatalyst, Mater. 2022, 131, 112591.
  2. Palanisamy, G.; Bhuvaneswari, K.; Bharathi, G.; Pazhanivel, T.; Dhanalakshmi, M. Improved photocatalytic performance of magnetically recoverable Bi2Te3/ CdS/CuFe2O4 nanocomposite for MB dye under visible light exposure, Solid State Sci. 2021, 115, 106584.
  3. Joorabi, F. T.; Kamali, M.; Sheibani, S. Effect of aqueous inorganic anions on the photocatalytic activity of CuO–Cu2O nanocomposite on MB and MO dyes degradation, Sci. Semicond. Process. 2022, 139, 106335.
  4. Kumar, S.; Kaushik, R. D.; Purohit, L. P. Novel ZnO tetrapod-reduced graphene oxide nanocomposites for enhanced photocatalytic degradation of phenolic compounds and MB dye, Mol. Liq. 2021, 327, 114814.
  5. Padmini, M.; Balaganapathi, T.; Thilakan, P. Rutile-TiO2: Post heat treatment and its influence on the photocatalytic degradation of MB dye, Int. 2022, 48, 16685.

Reviewer 2 Report

The manuscript titled " WSSe nanocomposite for enhanced photocatalytic hydrogen evolution and Methylene blue removal under visible-light irradiation"

In this manuscript, series of WS2@WSe2 nanocomposites with diverse Se mass ratio of 1wt%, 5wt% and 10wt% were synthesized and testified for the degradation of MB and photocatalytic hydrogen production. The accomplished photocatalyst performance is impressive. Therefore, I would like to recommend published this work after addressing the following points:

1. Authors should provide photocatalytic activities for hydrogen evolution with deviation (Fig. 5 and Fig 6c).

2. Some photoelectrochemical tests, such as EIS and LSV plots conducted under visible-light irradiation should be added.

3. They are different reactions of photodegradation of Methylene blue as well as H2 evolution. Thus, it is needed to confirm whether the activity enhancement mechanism is the same.

4. The reusability test is very interesting. Any possible explanation about the degradation reason? This might add extra insight to this paper

5. The comparison on photocatalytic activity and Hydrogen production of as-prepared samples with some typical tungsten-based photocatalysts ever reported should be added.

6. More information on the 420 nm filter should be added. If it is not a commercial one, the authors should provide lamp spectra before and after the insertion of the filter to show UV photons are not present.

7. The author claimed that the specific surface area increases significantly after combining with Se. Could the authors explain why the surface area increases?

8. The authors claimed that the total production of H2 was increased with increasing the Se ratio until reached maximum at WSSe-3 (10.0 wt.%), what about other weight percent after 10.0 wt.%.

9. Introduction part, if possible, some important and relative reports about Photocatalysis could helped: https://doi.org/10.1021/acsomega.1c03693 , https://doi.org/10.1016/j.jmrt.2022.03.067 , https://doi.org/10.1021/acsomega.1c03735

Hence, I recommend it accepted for publication after some major revisions.

Author Response

Journal: Materials (ISSN 1996-1944)

Manuscript ID: materials-1860580

Type: Article

Title: WSSe nanocomposite for enhanced photocatalytic hydrogen evolution and Methylene blue removal under visible-light irradiation

Author's Reply to the Review Report (Reviewer 2)

Comments and Suggestions for Authors

The manuscript titled " WSSe nanocomposite for enhanced photocatalytic hydrogen evolution and Methylene blue removal under visible-light irradiation"

In this manuscript, series of WS2@WSe2 nanocomposites with diverse Se mass ratio of 1wt%, 5wt% and 10wt% were synthesized and testified for the degradation of MB and photocatalytic hydrogen production. The accomplished photocatalyst performance is impressive. Therefore, I would like to recommend published this work after addressing the following points:

  1. Authors should provide photocatalytic activities for hydrogen evolution with deviation (Fig. 5 and Fig 6c).

Reply: Thank you very much for pointing out this matter. We have modified the Figure 4 and 5 with deviation in the revised manuscript.

  1. Some photoelectrochemical tests, such as EIS and LSV plots conducted under visible-light irradiation should be added.

Reply: Thank you very much for pointing out this. It’s true that our original statement is not clearly enough. According to your suggestion, we have added some additional description in the revised manuscript. It has been described in Page 10, Lines 458~469: “ Electrochemical impedance spectroscopy (EIS) was performed to check the transportation and charge carries separation rate of nanocomposite surface. 40 In Fig. 6f, the Nyquist semicircle radius of WSSe-3 is significantly smaller than that of pure WS2 and WSe2, verifying that the transfer resistance of photoexcited electrons and holes in WSSe-3 is the smallest among these catalysts. It was indicated that the WSSe-3 improved the charge transfer ability for composite. Firstly, we examined the linear sweeping voltammetry (LSV) curves of WS2, WSe2, WSSe-1, WSSe-2, and WSSe-3. in 0.1 M KOH solution at a scan rate of 2 mV s1 (Fig. 6g). Obviously, WSSe-3 nanocomposite displays the most greater LSV performance than other photocatalysts. These results further verify the lattice defects on WSSe-3 nanocomposite may be more useful for supplying more photocatalytic reaction sites, mainly enhancing redox activity area and further strengthening the MB degradation and hydrogen evolution action performance.”

Fig.6 (f) electrochemical impedance spectroscopy of the as-fabricated photocatalysts, and (g) LSV curves of the as-fabricated photocatalysts in 0.1 M KOH.

  1. They are different reactions of photodegradation of Methylene blue as well as H2 Thus, it is needed to confirm whether the activity enhancement mechanism is the same.

Reply: Thank you very much for pointing out this. According to the above measurement, the main process in H2-evolution were schematically illustrated in Fig. 7. The CB of WS2 was more negative than E (H+/H2), which make it possible for WS2 to conduct the H2-generation process. After the protonation process, the WSe2 exhibited higher separation efficiency of photo-excited carriers and higher H2-evolution ability due to the modified band structure with larger band gap energy and more negative CB position. In addition, the CB of WS2 is more negative than O2/O2, the e could be reacted with O2 molecules to produce O2 for decomposed MB solution. The photogenerated h+ in the VB of WSe2 could be directly reacted with MB solution or indirectly reacted with H2O to form OH for decomposed MB solution. The holes induced by photo irradiation was captured by the same photocatalytic mechanism with the electrons left in CB to finish the H2-generation process. With respect to WSSe nanocomposites, the charge carriers could separate more effectively due to the modified band structure and the enhanced surface area, which constitute the possible reason for the higher performance in H2 evolution. Owing to the intimate contact between WS2 and WSe2, efficient charge separation is achieved in WSSe nanocomposite photocatalyst, which accordingly boosts the photocatalytic performance towards H2 evolution and MB degradation under visible light illumination. This result was similar to the previous studies. 27, 30-33, 34-37 We have done support the statements and more specific on the introduction in the revised manuscript. It was modified in Page 12, Lines 554~556: “ This result of the photocatalytic performance towards H2 evolution and MB degradation under visible light illumination was similar to the previous studies. 27, 30-37

27 Mansha, M. S.; Iqbal, T. Experimental and theoretical study of novel rGO/AgVO3 nano-hetrostructures for their application as efficient photocatalyst, Opt. Mater. 2022, 131, 112591.

30 Nakayama, M.; Takeda, A.; Maruyama, H.; Kumbhar, V.; Crosnier, O. Cobalt-substituted iron-based wolframite synthesized via polyol route for efficient oxygen evolution reaction. Electrochem Commun 2020, 120, 106834.

31 Wang, C.;  Wang, R.;  Peng, Y.; Chen, J.; Li, J. Iron tungsten mixed composite as a robust oxygen evolution electrocatalyst. Chem Commun 2019, 55(73),10944.

32 Peng, Q.; He, Q.; Hu, Y.; TaylorIsimjan, T.; Hou, R.; Yang, X. Interface engineering of porous Fe2P-WO2.92 catalyst with oxygen vacancies for highly active and stable large-current oxygen evolution and overall water splitting. J. Energy Chem. 2022, 65,574.

33 He, J.; Zhou, X.; Xu, P.; Sun, J.; Promoting electrocatalytic water oxidation through tungsten-modulated oxygen vacancies on hierarchical FeNi-layered double hydroxide. Nano Energy 2021, 80, 105540.

34 Palanisamy, G.; Bhuvaneswari, K.; Bharathi, G.; Pazhanivel, T.; Dhanalakshmi, M. Improved photocatalytic performance of magnetically recoverable Bi2Te3/ CdS/CuFe2O4 nanocomposite for MB dye under visible light exposure, Solid State Sci. 2021, 115, 106584.

35 Joorabi, F. T.; Kamali, M.; Sheibani, S. Effect of aqueous inorganic anions on the photocatalytic activity of CuO–Cu2O nanocomposite on MB and MO dyes degradation, Mater. Sci. Semicond. Process. 2022, 139, 106335.

36 Kumar, S.; Kaushik, R. D.; Purohit, L. P. Novel ZnO tetrapod-reduced graphene oxide nanocomposites for enhanced photocatalytic degradation of phenolic compounds and MB dye, J. Mol. Liq. 2021, 327, 114814.

37 Padmini, M.; Balaganapathi, T.; Thilakan, P. Rutile-TiO2: Post heat treatment and its influence on the photocatalytic degradation of MB dye, Ceram. Int. 2022, 48, 16685.

  1. The reusability test is very interesting. Any possible explanation about the degradation reason? This might add extra insight to this paper

Reply: Thank you very much for pointing out this. The images of the preparation process have been added and support the statements as follow below. It has been described in Page 10, Lines 442~446: “ Seeing from the cycling experiment of WSSe-3 photocatalytic process, MB degradation rate slightly decreases due to the less photocorrosion caused by the remaining charges on the WSSe-3 surface. These results suggested that the fabricated WSSe-3 nanocomposite could be adopted for an excellent stability and repeatability during long-term photodecomposition system. ”

  1. The comparison on photocatalytic activity and Hydrogen production of as-prepared samples with some typical tungsten-based photocatalysts ever reported should be added.

Reply: Thanks for your suggestions. Photocatalytic process is an emerging, efficient and energy-save technology for the removal of organic contaminants and hydrogen evolution from the water environment. With the development of functional photocatalysts, nanocomposites has become one of the most popular emerging co-catalysts due to its high photocatalytic activity, strong adsorptivity, low cost and non-toxicity, especially applied to the photocatalytic degradation of organic contaminants. It has been described in Page 8, Lines 365~366: “ The performance differences of tungsten based photocatalysts for hydrogen evolution reaction as shown in Table 1. 30-33

Table 1. Studies of tungsten-based photocatalysts for the hydrogen production.

Photocatalyst

Tafel slope (mV/dec)

Overpotential at 10 mA/cm2

Exchange current density (mA/cm2 )

Electrolytes

reference

Co0.5Fe0.5WO4

36.3

360 mV

10

1.0 M KOH

30

Fe-WOx

51.7

0.38V

10

1.0 M KOH

31

Fe2P-WO2.92/NF

46.3

267mV

100

1.0 M KOH

32

FeNiW-LDH

55.7

202mV

10

1.0 M KOH

33

WSSe

65.2

252mV

10

0.1 M KOH

This work

30 Nakayama, M.; Takeda, A.; Maruyama, H.; Kumbhar, V.; Crosnier, O. Cobalt-substituted iron-based wolframite synthesized via polyol route for efficient oxygen evolution reaction. Electrochem Commun 2020, 120, 106834.

31 Wang, C.;  Wang, R.;  Peng, Y.; Chen, J.; Li, J. Iron tungsten mixed composite as a robust oxygen evolution electrocatalyst. Chem Commun 2019, 55(73),10944.

32 Peng, Q.; He, Q.; Hu, Y.; TaylorIsimjan, T.; Hou, R.; Yang, X. Interface engineering of porous Fe2P-WO2.92 catalyst with oxygen vacancies for highly active and stable large-current oxygen evolution and overall water splitting. J. Energy Chem. 2022, 65,574.

33 He, J.; Zhou, X.; Xu, P.; Sun, J.; Promoting electrocatalytic water oxidation through tungsten-modulated oxygen vacancies on hierarchical FeNi-layered double hydroxide. Nano Energy 2021, 80, 105540.

  1. More information on the 420 nm filter should be added. If it is not a commercial one, the authors should provide lamp spectra before and after the insertion of the filter to show UV photons are not present.

Reply: Thank you very much for pointing out this. The developed experiment on optical spectrometry allows the assessment of customary lamps with tungsten halogen lamp sockets. For that purpose, the irradiation at a certain position is collected with an optical fiber and analyzed with a USB compact spectrometer. The spectrometer used projects the received light through a 25 μm entrance slit onto a linear silicon CCD array with 651 active pixels offering approximately 2.0 nm optical resolution full width at half maximum. Radiation within a wavelength range of 300–750 nm is recorded with 12-bit A/D resolution. The photocatalytic activity under visible light irradiation (the 300-W Xe lamp) was tested with and without a cutoff filter to get rid of UV irradiation below 420 nm (Fig. R1).

Fig.R1 Spectral distribution of the 300-W Xe lamp with and without optical filter (λ≦420 nm).

  1. The author claimed that the specific surface area increases significantly after combining with Se. Could the authors explain why the surface area increases?

Reply: We appreciate the comments from this reviewer. We have been revised and added as follow above. It has been described in Page 5, Lines 219~227: “ The specific surface areas of WS2, WSe2, and WSSe-3 are 33.6, 23.3, and 67.4 m2 g-1, respectively, displaying that the specific surface area of WSSe-3 increases significantly after forming with nanocomposite. Nanocomposites have a large specific surface area and have a wide range of light absorption (i.e. from the ultraviolet to the visible region) for strong photocatalytic efficiency.23 In addition, the addition of selenium can adjust the surface electronic structure, change the surface morphology of the catalyst, and then affect the catalytic activity.24 The high surface area of the WSSe nanocomposites increases ion adsorption, resulting in increased ionic conductivity and specific capacitance of the material, in good line with previous report.25

23 Fan, Z.; Meng, F.; Zhang, M.; Wu, Z.; Sun, Z.; Li, A. Solvothermal synthesis of hierarchical TiO2 nanostructures with tunable morphology and enhanced photocatalytic activity, Appl. Surf. Sci. 2016, 360, 298.

24 Fang, G.Z.; Wang, Q.C.; Zhou, J.; Lei, Y.P.; Chen, Z.X.; Wang, Z.Q. Metal organic framework-templated synthesis of bimetallic selenides with rich phase boundaries for sodium-ion storage and oxygen evolution reaction. ACS Nano 2019,13, 5635.

  1. The authors claimed that the total production of H2 was increased with increasing the Se ratio until reached maximum at WSSe-3 (10.0 wt.%), what about other weight percent after 10.0 wt.%.

Reply: Thank you very much for pointing out this. According to our studied, when introducing Se into the WSSe matrix (Fig. R2), the H2-evolution variation tendency over the WSSe-4 heterostructures is similar to that of WSSe-3, and the WSSe-3 catalyst displays the optimal hydrogen generation efficiency (729.5 μmolh-1). Further increasing Se loading content leads to decreased photoactivity toward H2 production, which might associate with the “light-shielding effect” resulting from the excessive WSSe-3 cocatalyst. However, excessive loading of Se reduces the activity of H2 production, causing it to fall to almost zero. These results indicate that the Se is an effective cocatalyst for WSSe, and the loading ratio of Se needs to be controlled within an appropriate range. Among these, photocatalyst based on WSSe-3 nanocomposites has optimized chemical composition.  

Figure R2. Time courses of H2 production of WSSe nanocomposites.

  1. Introduction part, if possible, some important and relative reports about Photocatalysis could helped: https://doi.org/10.1021/acsomega.1c03693, https://doi.org/10.1016/j.jmrt.2022.03.067 , https://doi.org/10.1021/acsomega.1c03735

Hence, I recommend it accepted for publication after some major revisions.

Reply: Thank you very much for pointing out this. The number of references was expanded in the Introduction section by the addition of the following 3 references.

3 M. A.; Mannaa, K. F.; Qasim, F. T.; Alshorifi, S. M.; El-Bahy, R. S.; Salama, Role of NiO Nanoparticles in Enhancing Structure Properties of TiO2 and Its Applications in Photodegradation and Hydrogen Evolution, ACS Omega, 2021, 6, 30386.

4 El-Hakam, S. A.; ALShorifi, F. T.; Salama, R. S.; Gamal, S.; El-Yazeed, W. S. A.; Ibrahim, A. A.; Ahmed, A. I. Application of nanostructured mesoporous silica/ bismuth vanadate composite catalysts for the degradation of methylene blue and brilliant green, J. Mater. Res. Technol., 2022, 18, 1963.

5 Alshorifi, F. T.; Alswat, A. A.; Mannaa, M. A.; Alotaibi, M. T.; El-Bahy, S. M.; Salama, R. S. Facile and Green Synthesis of Silver Quantum Dots Immobilized onto a Polymeric CTS−PEO Blend for the Photocatalytic Degradation of p‑Nitrophenol, ACS Omega 2021, 6, 30432.

Round 2

Reviewer 2 Report

The manuscript is acceptable in the present form since authors have well addressed the questions proposed by referee